# Characterizing Highly Cited Papers in Mass Cytometry through H-Classics

**DOI:** 10.3390/biology10020104

**Published:** 2021-02-02

**Authors:** Daniel E. Di Zeo-Sánchez, Pablo Sánchez-Núñez, Camilla Stephens, M. Isabel Lucena

**Affiliations:** 1Servicio de Farmacología Clínica, Hospital Universitario Virgen de la Victoria, Instituto de Investigación Biomédica de Málaga (IBIMA), Facultad de Medicina, Universidad de Málaga, 29010 Malaga, Spain; cstephens@uma.es (C.S.); lucena@uma.es (M.I.L.); 2Departamento de Comunicación Audiovisual y Publicidad, Facultad de Ciencias de la Comunicación, Universidad de Málaga, 29010 Malaga, Spain; psancheznunez@uma.es; 3Centro de Investigación Social Aplicada (CISA), Edificio de Investigación Ada Byron, Universidad de Málaga, 29010 Malaga, Spain; 4Centro de Investigación Biomédica en Red de Enfermedades Hepáticas y Digestivas (CIBERehd), 28029 Madrid, Spain; 5UICEC IBIMA, Plataforma ISCiii de Investigación Clínica, 28020 Madrid, Spain

**Keywords:** mass cytometry, cytometry by time of flight (CyTOF), H-index, H-classics, highly cited papers (HCP), bibliometric indicators, scientometrics, science communication, Bibliometrix

## Abstract

**Simple Summary:**

The study of cell features has historically been key for the progress of biological sciences and its relevance remains intact. In recent years, mass cytometry has emerged as a promising and powerful technology, capable of studying multiple parameters of cells in the same sample. Mass cytometry has been quickly applied to many research areas, particularly to the study of the immune system for different purposes, in which the simultaneous analysis of a large number of proteins is crucial. However, despite being a technique that is on the rise, its performance in scientific publications has not yet been evaluated. In this work, a bibliometric methodology known as H-Classics was applied to analyse the most relevant articles, known as highly cited papers (HCPs), and determine the main scientific producers (authors, institutions, and countries) and trends around mass cytometry research field. The results confirmed a high interest and application in immunological studies. The identified HCPs came from prestigious institutions and were published in high impact journals. These results may help researchers to expand their knowledge and to establish new valuable collaborative networks around mass cytometry.

**Abstract:**

Mass cytometry (CyTOF) is a relatively novel technique for the multiparametric analysis of single-cell features with an increasing central role in cell biology, immunology, pharmacology, and biomedicine. This technique mixes the fundamentals of flow cytometry with mass spectrometry and is mainly used for in-depth studies of the immune system and diseases with a significant immune load, such as cancer, autoimmune diseases, and viral diseases like HIV or the recently emerged COVID-19, produced by the SARS-CoV-2 coronavirus. The objective of this study was to provide a useful insight into the evolution of the mass cytometry research field, revealing the knowledge structure (conceptual and social) and authors, countries, sources, documents, and organizations that have made the most significant contribution to its development. We retrieved 937 articles from the Web of Science (2010–2019), analysed 71 Highly Cited Papers (HCP) through the H-Classics methodology and computed the data by using Bibliometrix R package. HCP sources corresponded to high-impact journals, such as Nature Biotechnology and Cell, and its production was concentrated in the US, and specifically Stanford University, affiliation of the most relevant authors in the field. HCPs analysis confirmed great interest in the study of the immune system and complex data processing in the mass cytometry research field.

## 1. Introduction

Studying a broad range of single-cell features has always been of great interest to both researchers and clinicians. For decades, this purpose has been partly fulfilled by flow-cytometry technology [1]. For the study of cellular constituents and biomarkers by flow cytometry, cells are typically stained with monoclonal antibodies conjugated to fluorochromes. The key principle of flow cytometry is that, after the excitation of stained cells in a flow cytometer, a variety of aspects of individual particles can be measured through light scattering and fluorescence emission [2]. Flow cytometry has allowed the study of countless biological features [3] and is also being clinically used for the diagnosis of haematological diseases, such as leukaemia [4].

However, since flow cytometry is not without limitations, more comprehensive approaches to single-cell feature analysis have been developed, especially for multiparametric analyses [5]. Some examples are high dimensional fluorescent flow cytometry, single-cell mRNA sequencing, and mass cytometry [6]. The mass cytometry technology or cytometry by time of flight (CyTOF), is a relatively novel technique that was first described by Bandura et al., as an instrument for real-time analysis of individual biological cells or other microparticles [7]. The CyTOF constitutes a hybrid between flow cytometry and mass spectrometry. Instead of fluorophores, the cells are immunologically stained with antibodies conjugated with unique and stable metallic particles. Then, in the mass cytometer, samples ionised and focused under the same acceleration potential. The ions arrive at the detector in an ion mass-dependent manner, which enables the differentiation of single-cell parameters. For analysing this information, the detector is coupled to a time of flight analyser (TOF), from where the data is acquired and subsequently processed [8,9]. Due to the high availability of different metallic particles, mass cytometry allows the study of more than 40 different parameters simultaneously in the same sample [6].

The applications of mass cytometry are as broad as previous fluorescence-based techniques for studying single-cell features, with the exception that CyTOF does not allow cell recovery by sorting, as cells are destroyed in the process. Through performing deep profiling of samples, mass cytometry experiments have obtained comparable results to those of flow-cytometry, for example, being in the same way capable of quantifying surface and intracellular biomarkers from both non-stimulated [10] and stimulated [11] cells. The most important use of this technique is probably its capacity for immunophenotyping, that its, in-depth study of the characteristics and diversity of the immune system under different conditions [6]. Deep CyTOF profiling has also been applied in several drug pharmacodynamics, pharmacokinetics and discovery studies [12] to help determine better drug combinations and drug targets, especially for autoimmune diseases and cancer [13,14,15], and has also been used to support vaccine evaluation and development processes [16].

The huge potential of this technique suggests a booming trend in the use of CyTOF within the research community. However, limited information on the research evaluation of this new technology in scientific communications is currently available. A widely used approach for analysing scientific publications is the citation classics methodology. Citation classics was first defined by Garfield as a way to retrospectively determine the most highly cited papers (HCP) around a research topic [17]. The HCPs analysis aims to discover trends in the research community and identify relevant authors, institutions, or groups [17,18,19]. This analysis may facilitate comprehension of the research output and provide the basis for developing new theories, techniques, research lines and collaboration networks. However, to perform a good analysis, the criteria used to select the most cited papers should not be arbitrary. For example, some authors have arbitrarily established thresholds in the number of articles selected, limiting the list to the 50 [20] or 100 [21] most cited, or limiting the selection to those articles that have been cited at least 400 times [22].

To improve the selection criteria for HCP, Martinez et al. [23] proposed that the selection of HCPs should be based on two parameters: the H-Index [24] and the H-core concept [25]. The H-Index, defined as the number of papers with citation number > h, as was presented by Hirsch as a way to improve quantitative measures and impact of a researcher’s scientific output in one robust single parameter and avoid the disadvantage of using other indicators, such as the total number of papers, the total number of citations, citations per paper, or the number of significant papers [24]. Therefore, the application of H-index on citation classics reduces unpredictability [23].

Martinez et al. [23] gave the name H-Classics to this new identification method of HCPs based on the H-index. Two great advantages of using H-Classics is that it provides standardization of the criteria and thresholds used for selecting HCPs and includes the collection of papers published in a given field and their impact on a single procedure [26]. The H-Classics methodology has already been applied to medical areas such as paediatrics [27], dentistry [18], rheumatology [28], and microbiology [19].

In this paper, we focus on the study of mass cytometry research by using the H-Classics methodology. This H-Classics study aims to answer the following research questions (RQ):RQ1. Which are the most relevant journals, authors, institutions, and countries in mass cytometry research?RQ2. Which are the most cited documents in mass cytometry?RQ3. Which are the knowledge structures (conceptual and social structure) in mass cytometry?

## 2. Materials and Methods

### 2.1. Bibliographic Database and Query Design

The source of information related to scientific production and citations was the Web of Science (WoS) database (Clarivate Analytics). Academic publications indexed in WoS on CyTOF from 2010 to 2019 were obtained. To set and adequately delimit the research area under study, a specific search equation was formulated according to the database search logic of WoS. Table 1 demonstrates the query design, indexes, timespan, and data download date. The retrieved WoS dataset and the citation report are available at Zenodo repository [29].

### 2.2. H-Classics Methodology

After choosing the database and the query design, the following two steps for identifying HCPs were applied for the mass cytometry research field, accordingly to Martinez et al., description of the H- Classics methodology [23]:Compute the H-index of the research area. The computation of H-index of the research area is done by establishing a ranking of the papers according to their citations. The WoS database provides filtering tools to easily compute the H-index of the research area.Compute the H-core of the research area. This step consists of recovering the highly cited papers that are included in the H-core of the research area [23].

### 2.3. Bibliometrix (Science Mapping Analysis)

A comprehensive science mapping analysis [30,31] was performed to the data obtained from the HCPs selected. Data were processed and H-Classics scientific production was analysed through a bibliometric workflow supported by the open-source tool Bibliometrix package, which is programmed in R [32,33]. Following Aria et al., methodology, data were extracted from the Clarivate Analytics WoS database, loaded in R, and converted into a bibliographic data frame into which several elements, such as the authors’ names, titles, keywords, and other information, were introduced [33].

The descriptive analysis was performed applying several Bibliometrix functions described by Aria et al., to obtain tables and figures of HCPs production and citations over time, most productive authors, institutions, countries, most globally cited documents and authors production over time. Author productivity and citation impact were also calculated through H-index functions. The next step was the creation of networks in which different connections between attributes of a document were represented through a matrix. These connections were used to represent figures and networks corresponding to the collaboration network between HCPs’ authors (social structure) and co-occurrence network (conceptual structure) [33]. Co-ocurrence networks are generated by connecting pairs of terms (keywords) using a set of criteria defining co-occurrence. Data visualizations of author production and representation of social and conceptual structures were created through the Biblioshiny R package (https://bibliometrix.org/Biblioshiny.html), a Bibliometrix R Package web interface.

## 3. Results

### 3.1. Citation Report and Record Count 

The total mass cytometry publications retrieved (937) combined a sum of 25,801 times cited, making an average of 27.54 citations per paper. The H-index was 71, which means that 71 studies (Table A1) had received at least 71 citations and were therefore categorised as HCPs (H-Classics publications). Of these, 56 were original articles, 14 were reviews and 1 was a book chapter.

### 3.2. Distribution of Publications by Year, Average Citations per Year and Record Count

Figure 1 shows the yearly distribution of HCP documents from 2010 to 2019. The main production was concentred in the period 2014–2017. The peak number of HCPs occurred in 2016, with 17 HCP published. 

Figure 2 shows the average citations per year of HCPs in the mass cytometry field, remaining fairly constant between 2010 and 2018, before seeing a sharp increase to 208 in 2019.

### 3.3. Journals, Authors, Institutions and Countries

In this section, different social units (journals, authors, institutions, and countries) were analysed. First, the most productive journals were determined in terms of the total number of HCPs in the selected time frame. Table 2 shows journals with two or more HCPs in the mass cytometry research field. Nature Biotechnology with 9 HCPs was the most productive journal, followed by Cell and Cytometry Part A, with 8 and 5 HCPs, respectively.

Table 3 highlights the 20 most relevant authors in mass cytometry according to the number of published HCPs. Nolan GP ranked highest with 23 HCPs followed by Bendall SC (14 HCPs) and Newell EW (12 HCPs). Nolan GP was also one of the authors with the earliest date of first publication and the highest number of total citations (6531). The affiliations of these authors (75%) corresponded mainly to American hospitals, research centers, and universities.

Affiliations and countries were also analysed. To determine the most relevant institutions, all authors from each HCP were considered. Table 4 shows the ranking of institutions with 6 or more HCP. The most productive affiliations were compared with two quality and performance indicators of global university ranking: the 2019 Quacquarelli Symonds (QS) World University Rankings and 2019 Academic Ranking of World Universities (ARWU), which allow an approximate measurement of the relative position in which the most influential institutions in CyTOF publications were found. The most productive institution was Stanford University (USA), with 172 registered affiliations in HCPs, and it was also the affiliation of the top 3 authors (Table 3). As shown in Table 4, the 2019 QS/ARWU rankings indicate that top positions were held by Stanford University.

Table 5 shows the ranking of countries among producers of HCPs in mass cytometry, based on the affiliation of the corresponding author, and ordered according to the total number of HCPs. For a more global view of HCP contribution, the affiliations of all authors were also determined (Freq SCP) (Table 5). The countries of origin of all authors, and not only of the corresponding authors, were considered when determining the importance of a country in HCPs production.

Most of the H-Classics were derived from USA (*n* = 44, 61.97%), followed by Switzerland (*n* = 8, 11, 27%) and Singapore (*n* = 6, 8, 45%). Interestingly, only 25% of the USA HCPs were categorised as multiple country publications (MCP), the lowest ratio among the 12 top countries measured, after Canada and Germany. To relate gross domestic product (GDP) with the production of HCPs, Table 5 also displays the ranking of countries ordered by Adjustment Index (AI) based on GDP per capita [28]. The adjustment index (AI) was calculated as AI = ((total number of HCP/GDP per capita of the country) × 100). 

### 3.4. Content Analysis: Highly Cited Papers

The first HCP is from 2010, “Highly multiparametric analysis by mass cytometry” by Olga Ornatsky et al. [8]. They reviewed mass cytometry as a new technology capable of addressing the studies usually run by flow cytometers with better results, emphasizing its multiparametric analysis potential for biological research and drug development. To describe the basic principles of this technique, they used 20 different antibodies for immunophenotyping of human leukaemia cell lines and patient samples; and performed differential cell analysis, proteins identification, and metal-encoded bead arrays. This article pointed out the next steps in CyTOF development, such as improving measurement efficiency, the construction of isotope-binding polymer tags for a wider group of elements, and challenges of multiparametric data processing and representation. Ornatsky’s article, with 203 total citations, is the only HCP from 2010.

The 2011 article “Single-cell mass cytometry of differential immune and drug responses across a human hematopoietic continuum” by Bendall et al. [34] was the most cited mass cytometry HCP, with 1233 total citations (Table 6). In this article, healthy human bone marrow was used to simultaneously measure 34 parameters in single cells. It provided a system-wide view of immune signalling in healthy human haematopoiesis as a powerful way to compare mechanistic and pharmacologic studies. The second highest cited paper in the list (Table 6), with 716 total citations, was the 2012 article entitled “viSNE enables visualization of high dimensional single-cell data and reveals phenotypic heterogeneity of leukaemia” by Amir et al. In the article, the authors presented and validated a computerised tool that allows correct visualization of multiparametric mass cytometry information [35]. The third most cited paper was published by Ginsen et al., in 2014 and entitled “Highly multiplexed imaging of tumour tissues with subcellular resolution by mass cytometry” (500 total citations). The authors mixed mass cytometry with immunohistochemical techniques to study the heterogeneity of breast cancer tumours in tissues, rather than cell suspensions [36]. In 2011, Qiu et al. published the article “Extracting a cellular hierarchy from high-dimensional cytometry data with sPADE” [37], intending to solve some of the difficulties of analysing huge amounts of multidimensional single-cell data. The authors presented a computational approach that facilitates the analysis of cellular heterogeneity, the identification of cell types, and the comparison of functional markers in response to perturbations. This article was the fourth most cited HCP, with 498 total citations.

Finally, the latest HCP published was “Dimensionality reduction for visualizing single-cell data using UMAP” by Becht et al., (2019) [44], who used a nonlinear dimensionality-reduction technique for analysing mass cytometry biological data. This new algorithm, called UMAP, achieved a more efficient and improved visualization and understanding of single-cell data, which is still a challenge in recent years despite advances in computational methods for visualizing high dimensional data.

### 3.5. Top Author’s Production Over Time

Figure 3 shows the top authors’ production in mass cytometry HCPs over time. Authors such as Nolan GP or Bendall SC have maintained a consistent scientific production throughout almost the entire decade. Authors such as Simonds EF, Finck R, and Fantil WJ, had a higher contribution during the first half of the study period, while Spitzer MH had a more significant contribution during the second half.

### 3.6. Conceptual and Social Structure

#### 3.6.1. Conceptual Structure: Co-Occurrence Network

To analyse the frequency of keywords associated with the HCPs and study possible relationships between them, a co-occurrence keyword network was generated. Figure 4 shows the most used keywords in the analysed HCPs and predicted interconnections based on paired presence. Despite the presence of differentiated clusters, no noteworthy distinctions of nodes were observed. Frequent terms closely related to each other, such as “flow cytometry” and “mass cytometry”, were found to be interconnected with most other less frequent terms like “network” or “peripheral blood”. Terms such as “expression”, “T-cells, “macrophages”, “homeostasis” or “disease” tended to be interconnected. Similarly, terms like “responses”, “activation”, “populations”, “natural killer cells” were clustered together.

#### 3.6.2. Social Structure: Collaboration Network

Figure 5 shows the social structure through a collaboration network, where nodes are authors and links are co-authorships. There were two strongly differentiated clusters. On the one hand, the red cluster, led by Nolan GP and Bendall SC had a solid structure of collaborations with other influential authors, for example, Simonds EF, Finck R, and Levine JH. 

On the other hand, the violet cluster, led by Newell EW had a structure of collaboration with other impact authors such as Chen JM, Chan JKY, Ginhoux F and Mcgovern N among others. The author who acted as inter-collaborator between groups was Davis MM. The violet cluster appears related to a small orange cluster, with satellite collaborations involving Kurioka A and Kleneman P.

## 4. Discussion

In the present study, Mass Cytometry highly cited papers have been identified and analysed for the first time using the H-Classics methodology. The analysis of the HCPs allows us to highlight the following findings:

71 Mass Cytometry HCPs were identified in the period 2010–2019, with a predominance of original articles (56) compared to reviews (14). The increasing trend in citations per year presented in this work indicates a robust interest in mass cytometry in recent years, as CyTOF research is currently experiencing a potential growth phase. The HCPs were more frequent in the second half of the study period, with relatively recent publications attracting scientific interest. During the period analysed, 2016 showed the highest number of HCPs in mass cytometry. A peak so close to the end of the analysed time frame indicates that there is a growing interest in mass cytometry in the research community, considering that a minimum citation time window of 5 years is usually needed by a paper to obtain its maximum number of cites [45]. The immediate recognition of mass cytometry articles is probably also a consequence of most of them being published in open access journals, which usually results into a higher number of citations and impact compared to publications in non-open access journals [46]. Having this in mind and knowing that older articles are expected to have higher absolute citation rates, it is expected that the peak of HCPs will be actualised to a more recent date in years to come. 

Compared with other H-Classics studies on other disciplines, a smaller number of HCPs was identified in the present work. This is an expected result, considering that H-Classics studies usually focus on more general thematic areas and tend to analyse longer periods. For example, in the study by Moral-Muñoz et al., 645 HCPs in the microbiology area were analysed, covering a period of more than 100 years [19]. In our study, the timespan analysed was chosen considering that it covers most of the scientific production in mass cytometry. The rationale for choosing WoS as a source of the HCPs instead of other databases such as Scopus or Google Scholar, was that it provides numerous analysis tools for processing the data and offers highly reliable research information, which is particularly good for the study of biological sciences [47].

The HCP with the highest citations count, about differential immune and drug responses, was authored by Bendall et al., (2011). The reason that this article is the most cited in the field of mass cytometry may be due to two facts: first, it is one of the first works to successfully apply this novel technology (2011). Secondly, the paper highlights the utility of CyTOF in drug action and mechanistic studies and therefore could attract the attention of numerous pharmacology studies. The capacity of CyTOF for the evaluation of the effects of drugs both in the clinical and pre-clinical setting has been indicated in several studies [12,48,49]. The first HCP was published by Ornatsky et al., (2010) about immunophenotyping potential of the technique and it occupies position 24 out of 71. This article helps to understand how, from the beginning, this technique has been focused on the study of the immune system [8]. Interestingly, among the most cited papers, the second and the fourth-ranked were based on computer strategies to achieve adequate visualization and analysis of the results obtained by mass cytometry. This topic is repeated among HCPs throughout the entire decade studied, and was observed even in the most recently published HCP corresponding to 2019 [44]. The consistent interest in these articles suggests that the treatment of multiparametric data continues to be a challenge for this technique.

Nature Biotechnology was the most productive journal with 9 HCPs, followed by Cell with 8 HCPs. It is expected that the most cited articles of modern technology such as CyTOF, used in medicine and biology, will appear in more influential journals in this field. Nature Biotechnology is a journal belonging to Journal Citation Report from Web of Science, found in Q1 Quartile and ranked 02/156 (2019) in the area of Biotechnology & Applied Microbiology, with a Journal Impact Factor (JIF) of 36.56 (2019). High JIF journals are usually considered as a standard of quality and reliability in the scientific community. Equally applicable to our findings, high journal impact factors helped explain the high number of citations of HCPs studied in the field of paediatrics [27]. 

Nolan GP and Bendall SC were the authors present in the highest number of HCPs, 23 and 14, respectively. The high participation of these authors in HCPs consolidates them as world leaders in this technique and demonstrates the predominance of USA in global mass cytometry production, from which Standford University (172 affiliations registered) and Harvard University (28 affiliations registered) stand out as the most relevant institutions. According to ARWU/QS 2019, these institutions stand out for their high funding, internationalization of research, and quality of teaching staff/researchers. USA appears to have a low MCP ratio but, considering that the largest collaboration networks and the highest production of HCP occur in this territory, it is to be expected that the possibilities of international collaboration will be limited. This high scientific production is also possible thanks to a high number of collaborative works between the two main authors as they appear to cluster together in the collaboration network and have similar HCP production over time. Nolan GP leads the research group known as Nolan Lab (Stanford University), focused on the study of haematopoiesis, cancer and leukaemia, autoimmunity and inflammation, and computational approaches for network and systems immunology, with special knowledge in the mass cytometry technology [35].

Another collaboration cluster is that led by Newell EW, who is also the third author with regards to HCPs in mass cytometry, with 2044 total citations. Newell EW, affiliated to Stanford University & Singapore Immunology Network (SIGN), leads the Newell Lab research laboratory. This group innovatively applies mass cytometry to T-cell responses. They focus on unravelling the roles of extremely diverse immune cells with clinical implications in cancer and infectious diseases [50].

Although CyTOF has quickly emerged as a very useful and powerful technology for multidimensional analysis, the costs of the equipment and reagents (mass cytometer and metal-conjugated antibodies) are still much higher than those of conventional flow cytometry. This would explain why only groups and institutions with high funding and accessibility to a mass cytometer, such as Standford University, produce most of the HCPs in mass cytometry. The USA is the country with the highest funding in mass cytometry according to the Web of Science (Funding Agencies). American dominance in the production of H-Classics is also observed in other medical-related areas [27,28]. If we consider the value of AI as a strategy to adjust the production of HCPs to GDP, the ranking of the most productive countries in HCP would not change significantly. USA remains the country with the highest HCP and AI value (0.067). The only country for which the ranking would change is China. Despite having the lowest GDP in the list ($10,261.68), it has a considerable HCP production reflected in its AI (0.010).

Co-occurrence is a concept which refers to the common presence, frequency of occurrence, and close proximity of similar keywords associated with several articles [51]. Looking at the co-occurrence network (keywords) as an indicator of the conceptual structure of mass cytometry research field, keywords like “flow cytometry” and “mass cytometry” appear quite frequently linked and closely related to the rest of the network terms. The co-occurrence of these two terms was highly expected, as they are often complementary techniques or are under constant comparisons throughout the HCPs. These comparisons often highlight the advantages of one technology over the other. Probably the most mentioned issue is that flow cytometry has its multiplexing capacity limited by the number of markers (fluorochromes) that can be simultaneously analysed due to spectral overlap and autofluorescence [52], while that mass cytometry can accurately quantify more than 40 parameters on single cells due to the broadest spectrum of heavy-metals isotopes available [53].

Other topics that appear clustered together are those related to immune cells (“T-cells”, “macrophages”) or the ones related to the “activation” “responses” or “natural killer cells”. These frequently repeated topics were also expected among CyTOF HCPs as one of the most notable strengths of mass cytometry is its great immunophenotyping power for a vast diversity of study purposes. In the field of oncology research, the characterization of immune cell subsets through mass cytometry have pushed new knowledge in different types of cancer, such as melanoma, glioblastoma, pancreatic, breast or colon [54] and have also contributed to the study of less frequent diseases, such as the lungs Löfgren’s syndrome [55] or common ones like type I diabetes [56].

Other nodes that appear linked together are those related to data analysis (“differential expression analysis” or “network”). These results on the conceptual structure are further supported by what was observed in the content analysis mentioned above. The complexity of the multiparametric analysis involved in CyTOF has been reviewed in various publications [57,58]. 

It is important to highlight the impact that this technique is likely to have in areas such as clinical medicine, where it already has been proposed as a powerful tool for the diagnosis of autoimmune and complex hematologic diseases [59,60]. Similarly, a recent expansion of CyTOF, known as Imaging Mass Cytometry (IMC), has been proposed in the field of personalised medicine [61,62]. Finally, it is highly expected that CyTOF will contribute to the research on the recently emerged COVID-19 disease, caused by the SARS-COV2 coronavirus, which can cause serious immunological responses, in a similar way as it has contributed to the study of HIV [63]. In fact, this technology has already been used to elucidate the characteristics of the immune system in patients with COVID-19 during infection and recovery [64,65]. It would not be surprising if this technology gained more prominence in drug and vaccine development processes, as it has already supported the development and evaluation of hepatitis C and influenza vaccines [66,67].

## 5. Conclusions

In conclusion, this study shows for the first time the main protagonists and predominant trends in the area of mass cytometry, reflected in the most cited articles in the discipline. Likewise with studies in other areas, we think that this bibliometric information can help researchers, in particular those with an interest in immunology, to expand knowledge in this field and set valuable collaborations in the mass cytometry research field.

## Figures and Tables

**Figure 1 biology-10-00104-f001:**
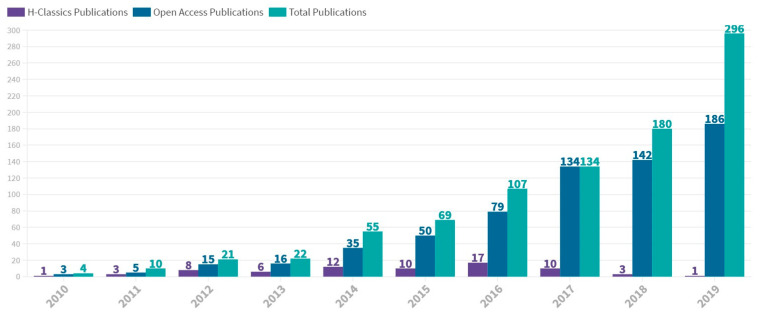
Annual scientific production in mass cytometry between 2010 and 2019. Number of H-Classics publications, open access publications and total publications are represented. The figure was designed with Flourish.

**Figure 2 biology-10-00104-f002:**
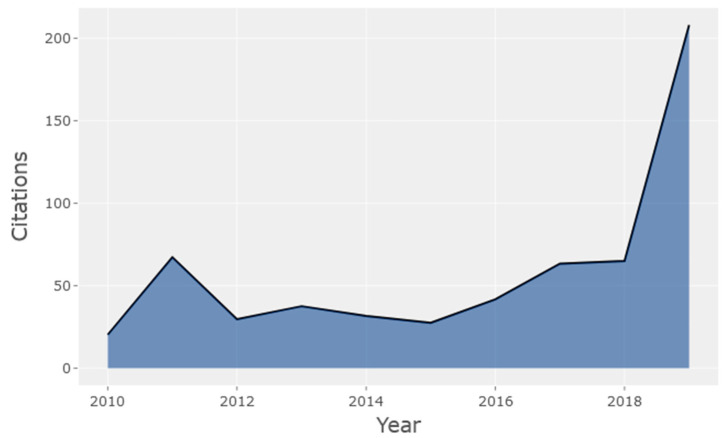
Average citations per year of HCPs in the mass cytometry research field (2010–2019).

**Figure 3 biology-10-00104-f003:**
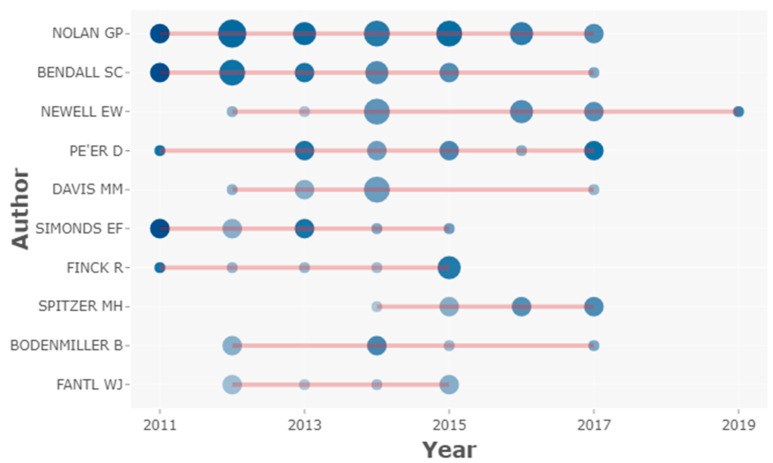
Top authors’ production over time. The horizontal red lines indicate the scientific production timeline per author from 2010 to 2019. The circle size represents the number of HCPs published per year and the colour intensity (colour coding) is proportional to the sum of total citations per year. The figure was created using Biblioshiny web interface.

**Figure 4 biology-10-00104-f004:**
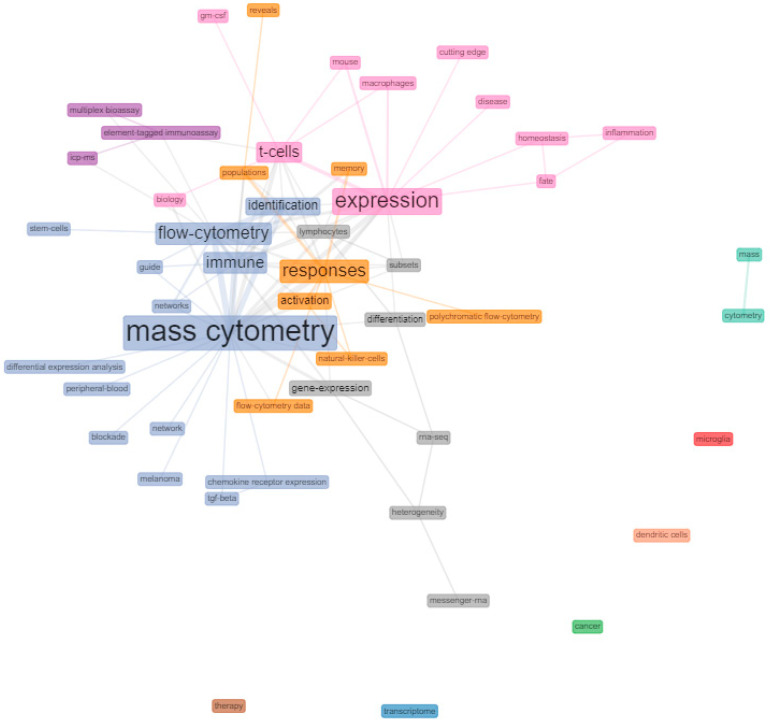
Keyword co-occurrence network of mass cytometry HCPs. Co-occurrence networks are shared interconnection of terms based on their paired presence within a definite unit of text. The size of the terms indicates their greater or lesser frequency with which they coincide with another term in the network throughout the HCPs (co-occurrence). Network parameters: field (keyword plus), network layout (Fruchterman & Reingold), normalization (association), node colour by year (No), clustering algorithm (Louvain), number of nodes (5–50), remove isolated nodes (No), and minimum edges (2). Graphical parameters: opacity (0–0.7), number of labels (0–50), label cex (Yes), label size (0–6), node shape (Box), edge size (0.1–5), and curved edges (No). The figure was created using Biblioshiny web interface.

**Figure 5 biology-10-00104-f005:**
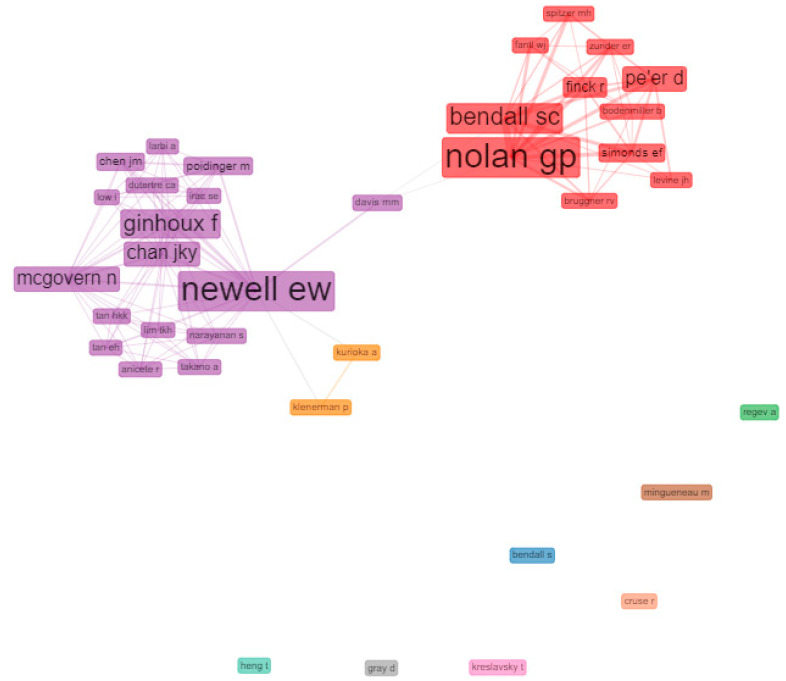
Clusters of most relevant author collaboration networks in mass cytometry’s HCPs. Network parameters: field (authors), normalization (No), clustering algorithm (Louvain), network Layout (automatic layout), number of nodes (37), remove isolated nodes (No), and minimum edges (2). Graphical parameters: opacity (0–0.7), number of labels (5–50), label cex (Yes), label size (0–6), node shape (Box), edge size (0.1–5), curved edges (No). The figure was created using Biblioshiny web interface.

**Table 1 biology-10-00104-t001:** Details of dataset search strategy for the identification of mass cytometry papers in WoS.

Indexes	Timespan	Query	N. of Documents	Download
Web of Science Core Collection: SCI-EXPANDED, SSCI, A&HCI, CPCI-S, CPCI-SSH, BKCI-S, BKCI-SSH, ESCI, CCR-EXPANDED, IC.	2010–2019	(TS = (“Mass Cytometry” OR “Cytometry by Time of Flight” OR “CyTOF”)) AND document types: (Article OR Review)	937	01.07.2020

**Table 2 biology-10-00104-t002:** Distribution of sources with two or more HCPs in the area of mass cytometry.

Rank	Sources	HCPs (%)	JIF 2019	JIF 5 Years	Quartile
#1	Nature Biotechnology	9 (12.68)	36.56	42.3	Q1
#2	Cell	8 (11.27)	38.64	38.62	Q1
#3	Cytometry Part A	5 (7.04)	3.12	3.47	Q2
#3	Immunity	5 (7.04)	22.55	25.73	Q1
#3	Science Translational Medicine	5 (7.04)	16.3	18.56	Q1
#4	Nature Immunology	4 (5.63)	20.48	22.3	Q1
#4	Science	4 (5.63)	41.85	44.37	Q1
#5	Nature Methods	3 (4.23)	30.82	36.15	Q1
#6	Cell Stem Cell	2 (2.82)	20.86	23.45	Q1
#6	Nature	2 (2.82)	42.78	46.47	Q1
#6	Nature Reviews Immunology	2 (2.82)	40.36	49.05	Q1
#6	Proceedings of The National Academy of Sciences of The United States of America	2 (2.82)	9.41	10.62	Q1
#6	Trends in Genetics	2 (2.82)	11.33	11.64	Q1

Abbreviations: JIF, journal impact factor.

**Table 3 biology-10-00104-t003:** Top 20 most relevant authors and their impact (H-index and TC) in the field of mass cytometry.

Rank	Authors	Articles	H- Index	TC	FPY	Affiliation	Country
#1	Nolan GP	23	23	6531	2011	Stanford University	USA
#2	Bendall SC	14	14	5197	2011	Stanford University	USA
#3	Newell EW	12	12	2044	2012	Stanford University & Singapore Immunology Network (Sign)	USA & Singapore
#4	Pe’er D	10	10	3978	2011	Columbia University	USA
#5	Simonds EF	8	8	3891	2011	Stanford University	USA
#5	Davis MM	8	8	1347	2012	Stanford University	USA
#6	Finck R	7	7	2626	2011	Stanford University	USA
#6	Spitzer MH	7	7	1155	2014	Stanford University & University of California	USA
#7	Bodenmiller B	6	6	1465	2012	Stanford University & University of Zurich	USA & Switzerland
#7	Zunder ER	6	6	1206	2012	Stanford University	USA
#7	Ginhoux F	6	6	1116	2014	Singapore Immunology Network (Sign)	Singapore
#7	Fantl WJ	6	6	759	2012	Stanford University	USA
#8	Bruggner EV	5	5	2356	2011	Stanford University	USA
#9	Levine JH	4	4	1641	2013	Columbia University	USA
#9	Davis KL	4	4	1591	2013	Stanford University	USA
#9	Krishnaswam YS	4	4	1225	2013	Yale School of Medicine	USA
#9	Chen JM	4	4	729	2014	Singapore Immunology Network (Sign)	Singapore
#9	Chan JKY	4	4	710	2016	Kk Women’s and Children’s Hospital & Duke-Nus Medical School	Singapore
#9	Mcgovern N	4	4	710	2016	University of Cambridge	UK
#9	Becher B	4	4	595	2014	University of Zurich	Switzerland

Abbreviations: TC, total citations; FPY, First publication year.

**Table 4 biology-10-00104-t004:** Top number of affiliations registered (institutions) in mass cytometry highly cited papers (HCPs).

Rank	Institutions	Country	AR	QS 2019	ARWU 2019
#1	Stanford University	USA	172	2	2
#2	Harvard University	USA	28	3	1
#2	University of Zurich	Switzerland	28	78	61
#3	Columbia University	USA	18	16	8
#3	Singapore General Hospital	Singapore	18	-	-
#3	University of Texas Md Anderson Cancer Centre	USA	18	-	68
#4	University of California, San Francisco	USA	16	-	20
#4	Weizmann Institute of Science	Israel	16	-	101–150
#5	National University of Singapore	Singapore	14	11	67
#6	KK Women’s and Children’s Hospital	Singapore	12	-	-
#6	Washington University	USA	12	100	22
#7	Howard Hughes Medical Institute	USA	10	-	-
#7	University of Oxford	UK	10	5	7
#8	Brigham and Women’s Hospital	USA	8	-	-
#8	Karolinska Institutet	Sweden	8	-	38
#8	University of Copenhagen	Denmark	8	79	26
#8	Universiteit Gent	Belgium	8	138	66
#8	University of Washington	USA	8	66	14
#8	Duke-NUS Medical School	USA	6	-	-

Abbreviations: AR, number of affiliations registered; QS, Quacquarelli Symonds; ARWU, Academic Ranking of World Universities.

**Table 5 biology-10-00104-t005:** Corresponding author’s country and country scientific production in the field of mass cytometry’s highly cited papers (HCPs).

Rank	Country	Freq CSP	HCPs (%)	SCP	MCP	MCP Ratio	Global Rank GDP	GDP/Capita ($)	AI
#1	USA	444	44 (61.97)	33	11	0.25	#8	65,280.68	0.067
#2	Switzerland	66	8 (11.27)	4	4	0.5	#4	81,993.73	0.01
#3	Singapore	104	6 (8.45)	2	4	0.667	#9	65,233.28	0.009
#4	United Kingdom	38	3 (4.23)	0	3	1	#23	42,300.27	0.007
#5	Belgium	18	2 (2.82)	1	1	0.5	#20	46,116.69	0.004
#6	Sweden	18	2 (2.82)	1	1	0.5	#14	51,610.07	0.004
#7	Canada	16	1 (1.41)	1	0	0	#19	46,194.72	0.002
#8	China	2	1 (1.41)	0	1	1	#68	10,261.68	0.01
#9	Denmark	14	1 (1.41)	0	1	1	#11	59,822.09	0.002
#10	Germany	18	1 (1.41)	1	0	0	#18	46,258.89	0.002
#11	Israel	18	1 (1.41)	0	1	1	#21	43,641.39	0.002
#12	Netherlands	12	1 (1.41)	0	1	1	#13	52,447.83	0.002

Multiple Country Publications (MCP) ratio was calculated dividing MCP by single Country Publications (SCP). Abbreviations: Freq CSP, Country scientific production Frequency; GDP, Gross Domestic Product; AI, Adjustment Index.

**Table 6 biology-10-00104-t006:** Top 10 most highly cited papers in mass cytometry (2010–2019).

Rank	N. Authors	Article	TC	TCY	Ref.
#1	16	Bendall SC, Simonds EF, Qiu P, et al. Single-cell mass cytometry of differential immune and drug responses across a human hematopoietic continuum. Science. 2011; 332(6030):687–696.	1233	123.3	[34]
#2	10	Amir el-AD, Davis KL, Tadmor MD, et al. viSNE enables visualization of high dimensional single-cell data and reveals phenotypic heterogeneity of leukaemia. Nat Biotechnol. 2013; 31(6):545–552.	716	89.5	[35]
#3	14	Giesen C, Wang HA, Schapiro D, et al. Highly multiplexed imaging of tumour tissues with subcellular resolution by mass cytometry. Nat Methods. 2014; 11(4):417–422.	500	71.4	[36]
#4	9	Qiu P, Simonds EF, Bendall SC, et al. Extracting a cellular hierarchy from high-dimensional cytometry data with SPADE. Nat Biotechnol. 2011; 29(10):886–891. Published 2 October 2011.	498	49.8	[37]
#5	3	Maecker HT, McCoy JP, Nussenblatt R. Standardizing immunophenotyping for the Human Immunology Project. Nat Rev Immunol. 2012; 12(3):191–200. Published 17 February 2012.	450	50	[38]
#6	1	Ransohoff RM. A polarizing question: do M1 and M2 microglia exist? Nat Neurosci. 2016; 19(8):987–991.	435	87	[39]
#7	16	Levine JH, Simonds EF, Bendall SC, et al. Data-Driven Phenotypic Dissection of AML Reveals Progenitor-like Cells that Correlate with Prognosis. Cell. 2015; 162(1):184–197.	413	68.8	[40]
#8	9	Bendall SC, Davis KL, Amir el-AD, et al. Single-cell trajectory detection uncovers progression and regulatory coordination in human B cell development. Cell. 2014; 157(3):714–725.	377	53.9	[41]
#9	5	Newell EW, Sigal N, Bendall SC, Nolan GP, Davis MM. Cytometry by time-of-flight shows combinatorial cytokine expression and virus-specific cell niches within a continuum of CD8+ T cell phenotypes. Immunity. 2012; 36(1):142–152.	360	40	[42]
#10	4	Bendall SC, Nolan GP, Roederer M, Chattopadhyay PK. A deep profiler’s guide to cytometry. Trends Immunol. 2012; 33(7):323–332.	351	39	[43]

Abbreviations: TC, total citations; TCY, total citation per year.

## Data Availability

The data presented in this study are openly available in Zenodo repository at http://doi.org/10.5281/zenodo.4462149.

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
