# Peer review of "Characterizing Highly Cited Papers in Mass Cytometry through H-Classics"

_biology, 2021, doi:10.3390/biology10020104_

Round 1
Reviewer 1 Report
The authors here carried out bibiloimetric analyses on mass cytometry publications from 2010 to 2019. Mass cytometry is indeed an important technology for deep phenotyping of the immune system and the manuscript described such importance well.
I cannot comment on the methods used in this manuscript as I am not familiar with bibliometric analysis methods. The manuscript is well-written, and the rationale is clear. I believe this manuscript would be of interest to the readers of Biology.
Author Response
We thank the reviewer for the very positive and encouraging comments on our work. The manuscript has undergone minor modifications to accommodate the recommendations of the editor and the other reviewers.
Reviewer 2 Report
The data from WOS was loaded and processed in R. The data and the R code developed for this analysis should be made available with a link to a persistent repository (i.e. DOI provided by the institution, Zenodo, etc.). This will greatly benefit the article and the scientific community.
Figure 1, the authors should consider increasing the font size to make the text easily readable.
The text for the description of Figure 2 “Figure 2 shows the average citations per year of HCPs in the mass cytometry field, indicating that the highest peak was reached in 2019 with 208 citations, following a rising trend” may read better “Figure 2 shows the average citations per year of HCPs in the mass cytometry field, remaining fairly constant between 2010 and 2018, before seeing a sharp increase to 208 in 2019.”
If the H-index is used to rank the authors in Table 3, the author with 5 articles and H-index should rank #8. In addition, the data in Table 3 should also be ranked using TCs (e.g. high to low for authors with the same H-index).
The authors should consider adding a legend to Figure 3 for the size and the colour coding of the circles.
The text “To analyse the frequency of key words associated” should read “To analyse the frequency of keywords associated”
The text in Figure 4 and Figure 5 overlap. The authors should also indicate how these figures were created. An alternative method to create such figures is with Gephi (open source); compatible with data export from R.
Author Response
We thank the reviewer for considering our manuscript and providing constructive feedback. We have addressed point by point all comments raised and generated a revised version of our manuscript titled "Characterizing Highly Cited Papers in Mass Cytometry through H-Classics", highlighting modified sections by using the “Track Changes" function in Microsoft Word. We think that the comments have contributed to improve the quality of our manuscript.
- The data from WOS was loaded and processed in R. The data and the R code developed for this analysis should be made available with a link to a persistent repository (i.e. DOI provided by the institution, Zenodo, etc.). This will greatly benefit the article and the scientific community.
Response: Thank you for your comment. The R code was not included as no specific code was developed or accessed manually for this work. The authors used the Biblioshiny graphical interface to represent the information obtained with bibliometrix. This information has been clarified in Material and methods section (2.3. Bibliometrix. Science Mapping Analysis, page 4). We have also included the description of the network parameters and the graphical parameters represented in the figures 4 and 5 (pages 13-16) of the knowledge structures (intellectual and conceptual), so it can be easily reproduced by other researchers.
Following your recommendation, we have included the dataset obtained from WoS, the dataset containing the H-classics and the citation report in a Zenodo repository, in case it might be necessary for other authors to reproduce or expand the analysis. The Zenodo repository has been referenced including its DOI in the “2.1. Bibliographic database and Query Design” section (page 3).
- Figure 1, the authors should consider increasing the font size to make the text easily readable.
Response: Thank you for your comment. Figure 1 has been modified according to the reviewer's proposed suggestions. The size of the body text of the axes, the legend and the labels has been expanded. In order to improve the data visualization, the font colours and the spacing and thickness of each column have also been modified.
- The text for the description of Figure 2 “Figure 2 shows the average citations per year of HCPs in the mass cytometry field, indicating that the highest peak was reached in 2019 with 208 citations, following a rising trend” may read better “Figure 2 shows the average citations per year of HCPs in the mass cytometry field, remaining fairly constant between 2010 and 2018, before seeing a sharp increase to 208 in 2019.”
Response: Thank you for your comment. The description text of Figure 2 has been modified according to the reviewer's proposed suggestions.
- If the H-index is used to rank the authors in Table 3, the author with 5 articles and H-index should rank #8. In addition, the data in Table 3 should also be ranked using TCs (e.g. high to low for authors with the same H-index).
Response: Thank you for your comment. We have corrected the position of the author that now occupies position #8 in Table 3, and in addition, we have ranked the authors who had the same H-index according to the TCs, as suggested.
- The authors should consider adding a legend to Figure 3 for the size and the colour coding of the circles.
Response: Thank you for this comment. The authors have rewritten the Figure 3 legend, clarifying the colour codes of the circles and lines.
- The text “To analyse the frequency of key words associated” should read “To analyse the frequency of keywords associated”
Response: Thank you for this comment. The text indicated by the reviewer has been replaced by the proposal.
- The text in Figure 4 and Figure 5 overlap. The authors should also indicate how these figures were created. An alternative method to create such figures is with Gephi (open source); compatible with data export from R.
Response: Thank you for this comment. We have redone Figures 4 and 5 using the Biblioshiny web interface to correct the overlap of terms and names.

Reviewer 3 Report
I congratulate the Authors of the paper, I find it very interesting. Mass cytometry is a novel technique and such a paper, for those readers who want to get acquainted with the mostly cited papers from the field. I truly recommend the paper to be printed in its current version.
Author Response
The authors thank the reviewer for the positive and encouraging comments on our manuscript. The manuscript has undergone minor modifications to accommodate the recommendations of the other reviewers.